# The Impact on Autonomic Nervous System Activity during and Following Exercise in Adults: A Meta-Regression Study and Trial Sequential Analysis

**DOI:** 10.3390/medicina60081223

**Published:** 2024-07-28

**Authors:** Jui-Kun Chiang, Yen-Chang Lin, Tzu-Ying Hung, Hsueh-Hsin Kao, Yee-Hsin Kao

**Affiliations:** 1Department of Family Medicine, Dalin Tzu Chi Hospital, Buddhist Tzu Chi Medical Foundation, No. 2, Minsheng Road, Dalin, Chiayi 622, Taiwan; roma@tzuchi.com.tw; 2Nature Dental Clinic, Puli Township, Nantou 545, Taiwan; drlin@alliswell.tw (Y.-C.L.); tzuyinghung@alliswell.tw (T.-Y.H.); 3Department of Radiation Oncology, Taichung Veterans General Hospital, Taichung 407, Taiwan; 4Department of Family Medicine, Tainan Municipal Hospital (Managed by Show Chwan Medical Care Corporation), 670 Chung-Te Road, Tainan 701, Taiwan

**Keywords:** exercise, sympathetic, parasympathetic, RMSSD, LF/HF ratio

## Abstract

*Background and Objectives*: Exercise enhances cardiovascular health through various mechanisms, including the modulation of autonomic nervous system activity. This study aimed to systematically examine the impact of exercise on heart rate variability (HRV) in adults during and within one hour after exercise (WHAE). *Materials and Methods*: A comprehensive literature review was conducted using the MEDLINE, Embase, Cochrane Library, Scopus, and PubMed databases to identify published studies that reported the impact of exercise on autonomic nervous system activity in adults. The studies measured the absolute power of the low-frequency band (0.04–0.15 Hz) to the absolute power of the high-frequency band (0.015–0.4 Hz) (LF/HF ratio) to assess sympathetic activity and the root mean square of successive differences between normal heartbeats (RMSSD) to assess parasympathetic activity. *Results*: A total of 3329 studies were screened for relevance, and finally, 10 articles that utilized methods for measuring autonomic nervous system activity, such as the LF/HF ratio and RMSSD, covering 292 adult patients, were included for meta-analysis. In the current meta-analysis, we observed a significant decrease in parasympathetic activity during and after exercise, as indicated by RMSSD, compared to pre-exercise levels (mean difference [MD] = −4.96, 95% confidence interval [CI]: −8.00 to −1.91, *p* = 0.003). However, sympathetic activity after exercise, represented by the LF/HF ratio, showed a borderline significant increase compared to pre-exercise levels (MD = 1.06, 95% CI: −0.01 to 2.12, *p* = 0.052). The meta-regression model found that factors associated with RMSSD included mean age, male gender, and duration post-exercise. Additionally, the factor associated with the LF/HF ratio was the healthy condition of participants. The trial sequential analysis provided robust evidence of a decrease in RMSSD and an increase in the LF/HF ratio during and WHAE. *Conclusions*: Given the limitations of the current study, the findings suggest that a significant decrease in parasympathetic activity and a borderline significant increase in sympathetic activity in adults during and WHAE, as confirmed by trial sequential analysis. Meta-regression analysis indicated that parasympathetic activity was negatively associated with participant age and male gender, but positively associated with duration post-exercise. Additionally, increased sympathetic activity was linked to the healthy conditions of participants. This study suggests that exercise might differentially affect autonomic balance in individuals with chronic conditions compared to healthy individuals. This highlights the potential need for tailored exercise interventions to improve autonomic function across different populations.

## 1. Introduction

Everyone can benefit from exercise, regardless of their health status or any existing diseases. Exercise involves engaging in physical activity to maintain or enhance health and fitness. Many individuals adopt a regular exercise regimen not only to improve their well-being, but also as a preventive or therapeutic measure to reduce the risk of chronic diseases and mortality [1]. In both disease prevention and treatment scenarios, physical activity and regular exercise contribute to a higher quality of life and potentially increased longevity [2]. A previous meta-analysis documented the favorable impact of physical activity on various health outcomes, including cardiovascular disease and all-cause mortality [3]. Additionally, exercise typically provides psychological benefits alongside physical fitness, both of which are associated with autonomic nervous function [4,5].

Exercise training plays a pivotal role as a therapeutic intervention for individuals with chronic stable heart failure or following myocardial infarction, highlighting its protective and rehabilitative effects [6]. A previous study also showed that nearly half of the patients with diabetes reached remission (non-diabetic state) and were able to stop taking their antidiabetic medications [7]. In patients with coronary artery disease (CVD), exercise training has been proven to enhance endothelium-dependent vasodilation, boost ejection fraction and exercise endurance, improve quality of life, and lower CVD-related mortality rates [8,9,10,11,12,13,14].

The autonomic nervous system, which regulates involuntary physiological processes such as heart rate, blood pressure, respiration, digestion, and sexual arousal, includes both sympathetic and parasympathetic activities. Parasympathetic activity is dominant while a person is at rest, whereas sympathetic activity is dominant during exercise. Exercise enhances cardiovascular health through various mechanisms, including heightened mitochondrial biogenesis and fatty acid oxidation [6,15,16,17], improved myocardial perfusion via vasodilation [18], reduced inflammation guarding against atherosclerosis development [19,20,21], and the modulation of autonomic nervous system activity, characterized by decreased sympathetic activity and increased parasympathetic tone [22,23]. Exercise training has been demonstrated to reduce muscle sympathetic nerve activity in various clinical populations with heightened sympathetic nervous system activity, including those with heart failure, metabolic syndrome, obesity, and hypertension [24,25,26,27,28]. A systematic review found that walking training led to increased parasympathetic modulation indices and/or decreased sympathetic modulation indices in patients with lower extremity arterial disease and symptoms of claudication [29]. Another study reported that healthy elderly participants show sympathetic predominance during exercise and do not return to baseline even after 30 min of recovery; similar responses were observed in stroke patients [30].

Various tests can assess autonomic nervous system activity, including measuring skin sympathetic nerve activity, conducting tilt table tests, and evaluating heart rate variability (HRV). HRV refers to the fluctuations in the time intervals between successive heartbeats [31]. It serves as an index of neurocardiac function, reflecting the interplay between the heart and the brain, as well as the dynamic and nonlinear processes of the autonomic nervous system (ANS) [32]. HRV is currently the most common method for assessing autonomic nervous system activity. HRV analysis includes both linear and nonlinear methods. Linear methods are categorized into two domains: time and frequency. These time-domain and frequency-domain methods are among the most common approaches for accurately assessing the function of the ANS [33]. Time-domain variables provide information on the distribution of time over the heart R-wave to R-wave (RR) time intervals, while frequency-domain variables provide insights into power distribution [34]. 

Due to significant divergence in study methodologies, there is a clear need for standardized tools to improve the quality of HRV measurements in current meta-analyses. A previous study identified the LF/HF ratio as a marker of sympatho-vagal balance, which indirectly reflects the level of cardiac sympathetic nervous system activity [35]. Some studies have used the high-frequency (HF) band in ms^2^/Hz as a proxy for parasympathetic tone, while the low-frequency (LF) band has often been used as a proxy for sympathetic activity [36]. In this study, we employed the LF/HF ratio, a frequency-domain measurement index, as an indicator of sympathetic activity, following the methodologies described in previous research by AlQatari et al. [37] and Kobayashi et al. [38]. Additionally, the RMSSD is used as a measure of parasympathetic activity, as referenced by the Task Force of the European Society of Cardiology and the North American Society of Pacing and Electrophysiology [39], Sequeira et al. [40], and Kobayashi et al. [41]. Resting HRV is affected by both psychological [42] and physiological factors, including age, gender, body mass index (BMI), chronic conditions [39], heart function, and heart diseases [43]. 

A previous systematic review reported that T2DM patients had significantly lower RMSSD compared to healthy participants, and LF/HF did not differ between T2DM and healthy groups [44]. In the current meta-analysis, our focus is exclusively on comparing HRV during exercise (including both during and 1 h after) with HRV before exercise. The dynamics of HRV are significantly altered during and after physical exercise due to the disruption of the balance between both branches of the ANS. Investigating the impact of exercise on the ANS was also a key objective of our study. Therefore, this study aimed to examine the impact on HRV during and within one hour after exercise (WHAE) in adults through a systematic review with meta-analysis and trial sequential analysis.

## 2. Materials and Methods

### 2.1. Study Design

A systematic review and meta-analysis were conducted to examine the effects of exercise on ANS activity in adults. The review adhered to the guidelines of the Preferred Reporting Items for Systematic Reviews and Meta-Analyses (PRISMA) [45]. The protocol for this systematic review was registered on PROSPERO (CRD42024557959). Additionally, the study protocol was reviewed and approved by the Research Ethics Committee of the Buddhist Dalin Tzu Chi Hospital in Taiwan (No. B11301022).

### 2.2. Search Strategy

We carried out a systematic literature search for English-language articles from Ovid MEDLINE, Embase, Cochrane Library, Scopus (Google Scholar), and PubMed, covering the period from inception to 19 May 2024 (Figure 1). Since 1985, terms such as physical activity, exercise, and training have often been confused with one another and are sometimes used interchangeably [46]. Physical activity in daily life can be categorized into occupational, sports, conditioning, household, or other activities. Exercise is a subset of physical activity that is planned, structured, and repetitive, with the final or intermediate objective of improving or maintaining physical fitness [47]. Training works by progressively increasing the force output of the muscles and involves a variety of exercises and types of equipment. We applied these keywords to search the candidate articles and then read them in detail. Finally, ten articles were associated with exercise and were included in this study. The following search strategy was developed for the PubMed/MEDLINE database: we utilized terms related to “exercise”, including variations such as “run*”, “walk*”, “jog*”, “treadmill*”, “tread mill*”, and “racewalk*”. These terms were searched within the title, abstract, and keywords. We systematically searched the Ovid, MEDLINE, Embase, Cochrane Library, Scopus, and PubMed for relevant articles. The candidate articles were initially selected, and duplicates were removed. In the current study, autonomic nervous system activity was measured by heart rate variability (HRV), with parasympathetic activity assessed by RMSSD and sympathetic activity assessed by the LF/HF ratio. The selection was based on the intersection of exercise with the parameters of RMSSD and the LF/HF ratio, resulting in a limit of 10 articles. Articles that did not include these measurements were excluded.

### 2.3. Study Selection

We used the PICOS (Participants, Interventions, Comparisons, Outcomes, and Study Design) model to frame our research question. The inclusion criteria were as follows: (1) Studies involving adult patients undergoing exercise; (2) Studies that employed methods to assess parasympathetic activity, such as RMSSD, and sympathetic activity, such as the LF/HF ratio, were considered during the data extraction process; and (3) Studies in which the activity of the autonomic nervous system was assessed during and within one hour after exercise (WHAE) in adults. Studies were excluded if the data were ambiguous, no post-exercise data were provided, or if communication with the corresponding authors was not possible. All searches were independently conducted by two reviewers (Y.-H.K. and J.-K.C.). Any disagreements between the reviewers were resolved through consultation with a statistician. Subsequently, these two authors further screened the papers and selected ten for analysis.

### 2.4. Data Extraction

The following data from the included studies were recorded: author, year of publication, sample size, age (years), gender (percentage of men), adult participants, and the values of parasympathetic activity (such as RMSSD) and sympathetic activity (such as the LF/HF ratio). The full information is summarized in Table 1. Data extraction was independently conducted by two reviewers (Y.-H.K. and J.-K.C.).

### 2.5. Methodological Quality of Included Studies

After acquiring the eligible articles, data extraction and methodological quality assessment were performed by two independent reviewers (Y.-H.K. and J.-K.C.). Methodological quality was assessed using the Downs and Black Checklist (Appendix A) [48], which includes 27 items across five subscales: reporting, external validity, internal validity (study bias and confounding), selection bias, and study power. Scores are categorized as follows: poor quality (14 or less), fair quality (15 to 19), good quality (20 to 25), and excellent quality (26 or higher) [49].

### 2.6. Risk of Bias of Included Studies

After extracting and critically appraising the data, we analyzed the included papers through narrative synthesis. To aid this process, we referred to the flowchart by Rodgers et al. [50], which offers guidance on synthesizing data, identifying patterns, evaluating the strength of these patterns, and drawing conclusions. The risk of bias was then evaluated using the ROBINS-I criteria from Cochrane.

### 2.7. Statistical Analysis

The objective of the current meta-analysis was to determine the weighted average of HRV mean differences, which were presented as the pooled summary effect in the forest plot. Heterogeneity among the included studies was assessed using the chi-square Q test and the I^2^ statistic. 

This model employed the weighted least squares method to identify relevant covariates, termed “moderators” in meta-analysis, which contributed to the observed heterogeneity. The moderators considered in the study included mean age, sex (male, female, and mixed samples), the percentage of males, and the time elapsed after exercise (in minutes). Additionally, if substantial residual heterogeneity persisted after the initial analysis, a mixed-effects linear meta-regression model for HRV mean differences was employed. 

To ensure the reliability and validity of our mixed-effects linear meta-regression analysis of HRV mean difference, we followed essential model fitting techniques: (1) Variable Selection; (2) Goodness-of-Fit (GOF) Assessment, such as calculating the coefficient of determination (R^2^); and (3) Regression Diagnostics, including residual analysis, detection of influential studies, and sensitivity analyses. To identify potential publication bias, we performed a funnel plot analysis. Specifically, Egger’s test was used to assess the symmetry of the funnel plot. For dealing the heterogeneity, we had checked again the data. A random-effects meta-analysis was used to incorporate heterogeneity among studies. Heterogeneity was explored by conducting subgroup analyses or meta-regression. In general, it is unwise to exclude studies from a meta-analysis on the basis of their results as this may introduce bias.

Trial sequential analysis (TSA) was used to assess whether the sample size was sufficient, based on a type I error rate of 5% and a power of 90%. This method establishes boundaries for benefit, harm, and futility, and evaluates overall efficacy through cumulative z-scores. This post hoc approach enabled us to determine whether the data provided convincing evidence of the true effect.

Meta-analysis, meta-regression, and trial sequential analyses were conducted by the R statistical software (version 4.2.3, provided by the R Foundation for Statistical Computing, Vienna, Austria). We used Cochrane’s Review Manager software (RevMan 5.4) to plot the risks of bias. A significance level of *p* ≤ 0.05 (two-sided) was considered statistically significant, ensuring rigorous interpretation of our findings.

## 3. Results

A total of 3329 studies were screened for relevance. Among these, 1040 studies were excluded due to duplication, and 710 studies were excluded as they did not meet the criteria of PICO. Additionally, 1575 full-text articles were excluded because they lacked methods and criteria for measuring autonomic nervous system activity, such as the LF/HF ratio and RMSSD. Figure 1 provides an overview of the study selection process. Ultimately, 10 articles covering 292 adult patients were included in the meta-analysis, after excluding articles lacking HRV data from the corresponding exercise sessions. The studies, published between 2014 and 2024, were evaluated for methodological quality utilizing the Downs and Black quality assessment method. All articles were classified as fair. Details of these 10 studies are presented in Table 1. The risk of bias assessment was conducted using the Cochrane tool, resulting in (a) a risk of bias graph and (b) a risk of bias summary (Figure 2). According to the Downs and Black Checklist, we found that the study quality was fair, with a mean score of 17. Most of the articles did not meet the criteria for internal validity, such as making attempts to blind study subjects to the intervention and making attempts to blind those measuring the main outcomes of the intervention. Additionally, the articles did not meet the criteria for external validity, such as ensuring that the subjects asked to participate in the study were representative of the entire population from which they were recruited, and that the staff, places, and facilities where the patients were treated were representative of the treatment the majority of patients receive.

In the 10 articles we analyzed, the term “exercise” was used in these topics. To standardize terminology related to exercise, physical activity, and training, “exercise” will be used consistently throughout the current study. These articles included both experimental and control groups, incorporating various exercise modalities such as aerobic exercise, resistance exercise, table tennis, yoga, cycling, and treadmill walking. In the current study, we divided these 10 studies into two groups: a healthy group [30,51,52,53,54] and a chronic conditions group [30,55,56,57,58,59], which included participants with T2DM, hypertension, and stroke. In total, the 10 articles comprised 56 subgroups (31 sympathetic subgroups and 25 parasympathetic subgroups). We further compared healthy participants with those who had chronic conditions such as T2DM, hypertension, and stroke in terms of RMSSD and LF/HF ratio during and WHAE in adults. 

In the current meta-analysis, the total forest plot analysis showed a significant decrease in parasympathetic activity during and WHAE, as indicated by RMSSD, compared to pre-exercise levels (mean difference [MD] = −4.96, 95% confidence interval [CI]: −8.00–−1.91, *p* = 0.003). Further subgroup analysis revealed that RMSSD values decreased mainly in participants with healthy conditions (MD = −5.35, 95% CI: −9.43–−1.26, *p* = 0.014) (Figure 3). The Egger’s test for RMSSD reached statistical significance (*p* = 0.001), possibly indicating an effect from the small number of collected articles. Another possible reason could be that the journal did not accept the negative results. The funnel plot for RMSSD is shown in Figure 4a. In the meta-regression model for RMSSD, the significant factors included mean age (β = −0.10, 95% CI: −0.18–−0.03, *p* = 0.006), male gender (β = −10.64, 95% CI: −17.19–−4.08, *p* = 0.002), and duration after exercise (β = 0.10, 95% CI: 0.02–0.18, *p* = 0.019) (Table 2). 

In the current meta-analysis, sympathetic activity after exercise, represented by the LF/HF ratio, showed a borderline significant increase compared to pre-exercise levels (MD = 1.06, 95% CI: −0.01–2.12, *p* = 0.052). Subgroup analysis revealed that the LF/HF ratio values were not significantly different between participants with healthy conditions and those with chronic conditions (Figure 5). The Egger’s test did not detect funnel plot asymmetry (*p* = 0.571). The funnel plot for the LF/HF ratio is shown in Figure 4b. In the meta-regression model for the LF/HF ratio, the significant factor was participants with healthy conditions compared with participants with chronic conditions (β = 1.45, 95% CI: 0.005–2.90, *p* = 0.049) (Table 2). 

Using trial sequential analysis, we confirmed that parasympathetic activity decreased while sympathetic activity increased during and WHAE in adults (Figure 6). 

Regarding sensitivity analyses, the conclusions remained consistent both before and after excluding each individual study. In the RMSSD analysis, the results were similar after removing the extreme data points (Colberg, 2014, table tennis [58]) compared to when the extreme data points were included. Similarly, in the LF/HF ratio analysis, after removing the extreme data points [51]), the results exhibited the same pattern as when the extreme data points were included.

## 4. Discussion

In our current meta-analysis, we observed that the acute effects of exercise might include an increased heart rate and cardiac output, primarily due to a significant withdrawal of parasympathetic tone and a borderline significant increase in the activation of the sympathetic nervous system in adults, both during and within one hour after exercise. The decrease in parasympathetic activity was primarily observed in the healthy group. Meta-regression analysis revealed that the withdrawal of parasympathetic nervous system activity was negatively associated with mean age and male gender, and positively associated with the duration after exercise. Additionally, the analysis showed that the increase in sympathetic activity was more pronounced in participants in healthy conditions compared to those with chronic conditions. Trial sequential analysis provided strong evidence of a decrease in RMSSD and an increase in the LF/HF ratio during and WHAE in adults.

The acute effects of exercise include an increased heart rate and cardiac output. The potential mechanisms linking exercise to autonomic function might be as follows. Parasympathetic activity is dominant while a person is at rest. During exercise, a significant withdrawal of parasympathetic tone and a borderline significant increase in the activation of the sympathetic nervous system occur in adults, both during and within one hour after exercise. During the recovery from exercise, parasympathetic activity gradually increases and sympathetic activity decreases to the baseline rest status. 

A previous systematic review reported that, in the absence of exercise, T2DM patients had significantly lower RMSSD compared to healthy participants, while LF/HF did not differ between T2DM and healthy groups [44]. In the current study, we found that participants with chronic conditions showed an increasing trend in sympathetic activity, as indicated by the LF/HF ratio, while parasympathetic activity, measured by RMSSD, did not significantly decrease during or within one hour after exercise. This suggests that exercise might differentially affect autonomic balance in individuals with chronic conditions, potentially highlighting the need for tailored exercise interventions to manage autonomic function in this population.

A previous study reported that parasympathetic reactivation plays a greater role later in recovery after exercise [60]. This is supported by the observation that plasma norepinephrine peaks approximately one minute after high-intensity exercise, indicating that sympathetic activity remains elevated during the early stages of recovery [61]. Therefore, the early decrease in heart rate during recovery is highly dependent on parasympathetic reactivation [62]. Another earlier study reported that following the cessation of exercise, parasympathetic activity exhibits a time-dependent recovery and eventually returns to pre-exercise levels [63]. In the current study, we found that the mean age of participants and their gender influenced parasympathetic activity, as indicated by RMSSD, after exercise. For example, according to the results of the current study, a 30-year-old male participant needs 176 min (−0.10 × 30 + (−10.64) + 0.10 × X minutes = 4.03) to return the RMSSD to the pre-exercise baseline level. Another example shows that a 30-year-old female participant needs 70 min (−0.10 × 30 + 0.10 × X minutes = 4.03) to return to the pre-exercise baseline. 

Additionally, the change in the LF/HF ratio, reflecting sympathetic activity, showed a borderline significant increase after exercise and was not significantly different between the healthy and chronic condition groups in the current meta-analysis. The meta-regression analysis revealed that sympathetic activity increased more in participants with healthy conditions compared to those with chronic conditions. 

A previous systematic review reported that endurance training and high-intensity interval exercise (HIIE) improved both sympathetic activity, as indicated by the LF/HF ratio, and parasympathetic activity, as indicated by RMSSD, in patients with T2DM. Additionally, resistance training improved sympathetic activity alone, while a combination of endurance and resistance training did not result in significant changes in either RMSSD or the LF/HF ratio after a regular exercise program [64]. In the current study, participants engaged in similar activities, including aerobic exercise, resistance exercise, and treadmill walking, across both the healthy and chronic condition groups. However, there were specific differences between the groups. In the healthy group, exercises included using a cycle ergometer to compare mild and moderate intensities, as well as passive recovery versus cool-down. In the chronic disease group, activities included high-intensity interval exercise (HIIE), moderate-intensity continuous exercise (MICE), table tennis, and yoga. The current study investigates the impact of exercise on autonomic nervous system activity both during exercise and within 60 min post-exercise. One explanation for the observed differences may be the influence of various factors on sympathetic nervous system activity. A previous review reported that although HRV is not widely accepted as a reflection of sympathetic activity, cardiac sympathetic activity may instead be investigated using systolic time intervals, such as the pre-ejection period [65]. Another explanation might be insufficient exercise duration or type, or an inadequate sample size. Further investigations are warranted to explore these factors, as they represent a limitation of the current study. 

A previous systematic review explored the long-term effects of exercise on HRV, noting improvements in all HRV parameters following exercise training. The study durations ranged from 1 to 9 months in individuals with T2DM, with findings including increases in RMSSD and decreases in the LF/HF ratio [64]. However, the current study aims to investigate the impact of exercise on HRV during and within one hour after exercise in adults, using a systematic review with meta-analysis and trial sequential analysis. A previous study reported that regular exercise for more than 2 months could improve the baseline rest parasympathetic activity [66]. However, the definite timing and real mechanisms of linking exercise to autonomic nervous system activity changes warrant further study to clarify. 

In the current study, we analyzed the diverse exercise outcomes of participants under various conditions, treating them as separate subgroups for analysis. Among these articles, only Parks et al. [53] and Sun et al. [54] involved young adults as the research subjects, while the remaining articles focused on the elderly population. Whether age is a factor influencing the effect of exercise on autonomic nervous system activity requires further research for clarification.

There are several limitations in the current study. Firstly, due to the ephemeral nature of sympathetic nervous system expression, these articles present sympathetic nervous system activity in terms of mean values. The immediate surge in sympathetic nervous system activity during exercise may not always be promptly captured. Future research utilizing real-time HRV testing (e.g., 300 s windows shifting by 10 s) to assess sympathetic nervous system activity may provide a more accurate representation of the impact of exercise. Secondly, the recovery of parasympathetic activity from exercise is influenced by exercise modality [67,68], duration [63,67], intensity [63], age, and gender. The effect of exercise intensity and duration needed to improve autonomic nervous system activity still warrants further investigation. However, not all of these factors were provided in the articles included in the current meta-regression study. Further investigation is warranted to explore these aspects. Thirdly, exercise adherence or participant characteristics like fitness level were not full available in the collected articles. Fourthly, confounding factors included age (ranging from 23.0 to 71.1 years, and from 51.0 to 71.1 years for those with chronic conditions), BMI (ranging from 22.5 to 34.8 kg/m^2^, and from 24.2 to 34.8 kg/m^2^ for those with chronic conditions), types of exercise (with resistance exercise, cycle ergometer, and treadmill being the most common), race, and other variables. Consequently, the findings of this study may not be generalizable to individuals outside these ranges. Fifthly, the potential biases in the current study included those related to obtaining individual participant data, publication bias, selection bias, and availability bias. Finally, there is considerable methodological variation in the literature concerning HRV responses to exercise, including differences in exercise protocols and HRV analysis techniques. This highlights the need for further research to standardize methods and ensure consistency in exercise protocols.

## 5. Conclusions

In the current meta-analysis, we observed a decrease in parasympathetic activity and a borderline significant increase in sympathetic activity in adults during and WHAE. The decrease in parasympathetic activity was primarily observed in healthy participants. A meta-regression analysis revealed that parasympathetic activity was negatively influenced by participant age and male gender, and positively influenced by the duration after exercise. Additionally, sympathetic activity increased and was associated with the healthy condition of the participants. The impact of exercise (during exercise and one hour afterward) on autonomic nervous system activity differed between participants with healthy conditions and those with chronic conditions. Trial sequential analysis provided strong evidence of a decrease in RMSSD and an increase in the LF/HF ratio during and WHAE in adults. The acute effects of exercise might include an increased heart rate and cardiac output, primarily due to a significant withdrawal of parasympathetic tone and a borderline significant increase in the activation of the sympathetic nervous system in adults, both during and within one hour after exercise. This study suggests that exercise might differentially affect autonomic balance in individuals with chronic conditions compared to healthy individuals. This highlights the potential need for tailored exercise interventions to improve autonomic function across different populations.

## Figures and Tables

**Figure 1 medicina-60-01223-f001:**
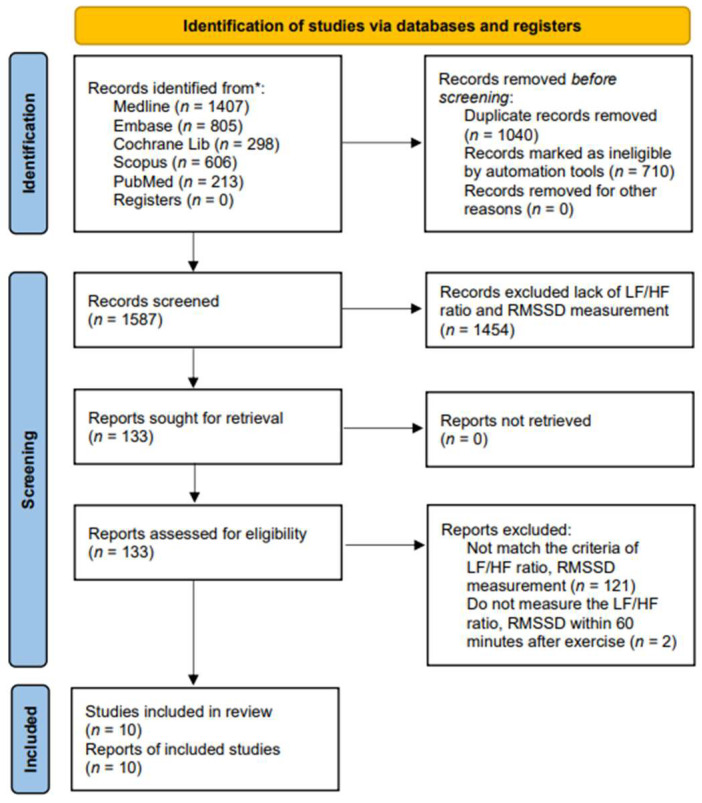
PRISMA flow diagram of the study selection. * Each database along with the corresponding numbers of records identified.

**Figure 2 medicina-60-01223-f002:**
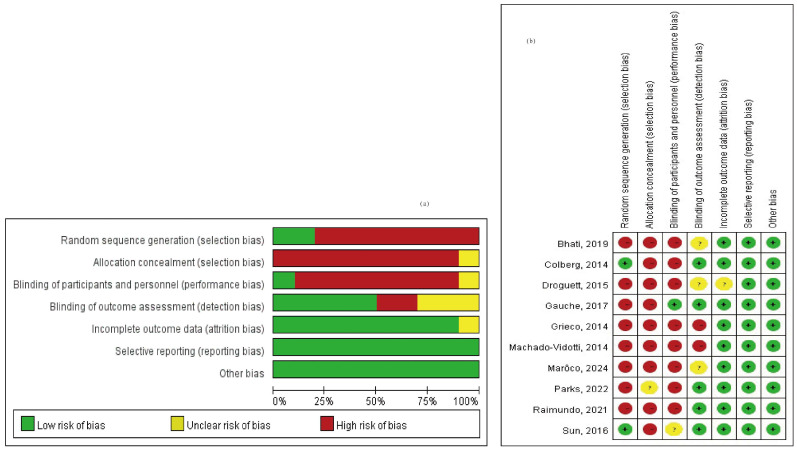
Risk of bias assessment (**a**) risk of bias graph and (**b**) risk of bias summary [30,51,52,53,54,55,56,57,58,59].

**Figure 3 medicina-60-01223-f003:**
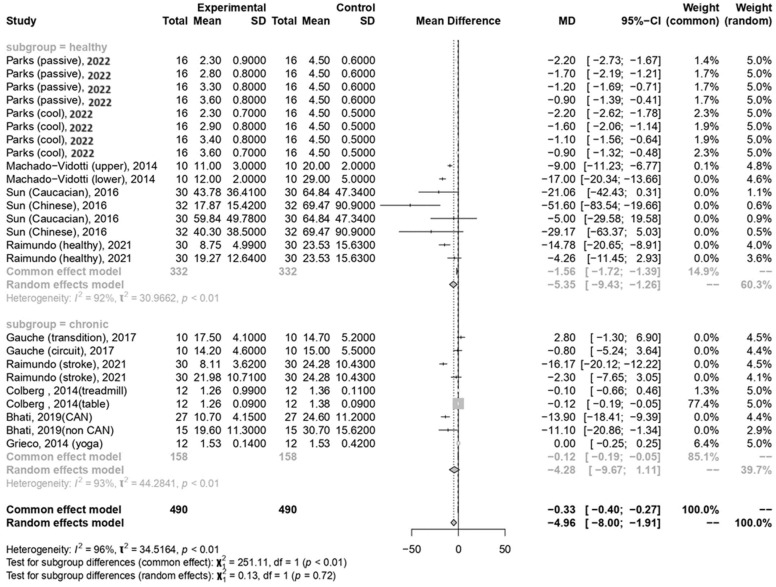
Summary of subgroup analysis (healthy and chronic counterparts) on RMSSD values [30,52,53,54,55,57,58,59].

**Figure 4 medicina-60-01223-f004:**
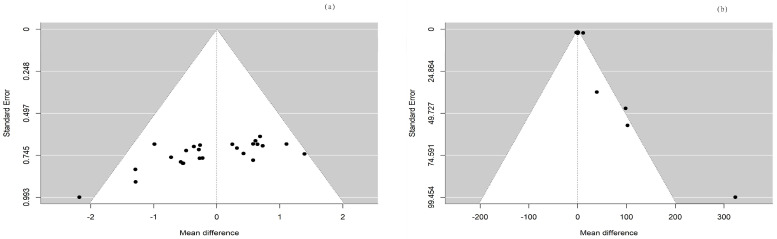
The funnel plots for (**a**) RMSSD and (**b**) LF/HF ratio.

**Figure 5 medicina-60-01223-f005:**
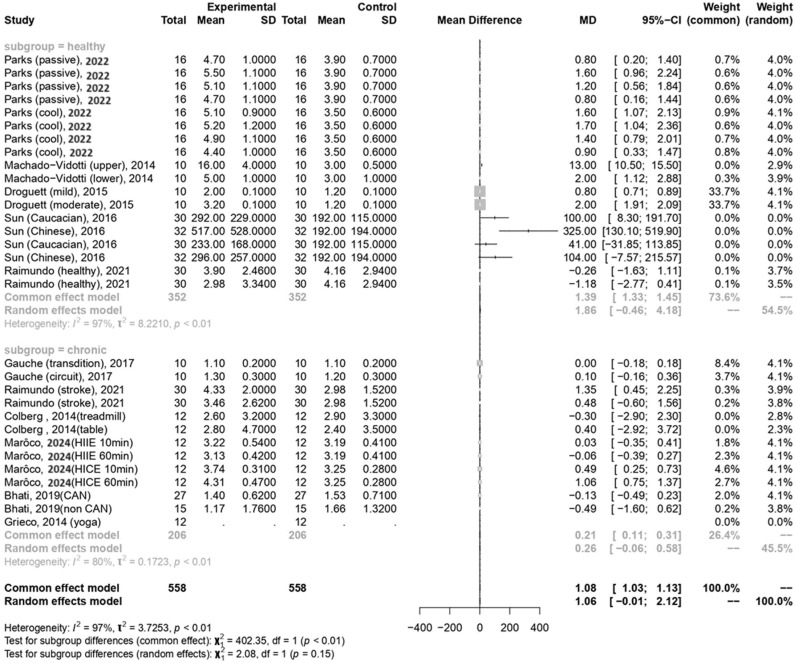
Summary of subgroup analysis (healthy and chronic counterparts) on LF/HF ratio [30,51,52,53,54,55,56,57,58,59].

**Figure 6 medicina-60-01223-f006:**
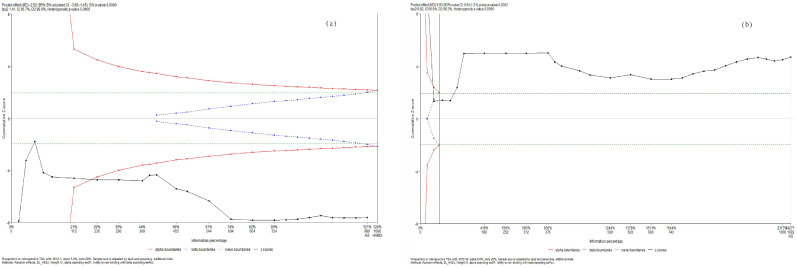
Trial sequential analysis for (**a**) RMSSD and (**b**) LF/HF ratio.

**Table 1 medicina-60-01223-t001:** Characteristics of studies included in this systematic review.

Publication	Types of Exercise	Health Status	N, Age (Years Old)	Sympathetic Activity, Measured by the LF/HF Ratio	Parasympathetic Activity, Measured by the RMSSD
Raimundo, 2021 [30]	Aerobic exercise	Healthy	N = 30, 67 ± 4	Pre-exercise: 4.16 ± 2.94 During exercise: 3.90 ± 2.46 30 min post-exercise: 2.98 ± 3.34	Pre-exercise: 23.53 ± 15.63 During exercise: 8.75 ± 4.99 30 min post-exercise: 19.27 ± 12.64
Raimundo, 2021 [30]	Aerobic exercise	Stroke	N = 30, 69 ± 3	Pre-exercise: 2.98 ± 1.52 During exercise: 4.33 ± 2.00 30 min post-exercise: 3.46 ± 2.62	Pre-exercise: 24.28 ± 10.43 During exercise: 8.11 ± 3.62 30 min post-exercise: 21.98 ± 10.71
Gauche, 2017 [55]	Traditional resistance exercise	Elderly women with hypertension	N = 10, 71.1 ± 5.5	Pre-exercise: 1.10 ± 0.20 60 min post-exercise: 1.10 ± 0.20	Pre-exercise: 14.70 ± 5.20 60 min post-exercise: 17.50 ± 4.10
Gauche, 2017 [55]	Circuit-based resistance exercise	Elderly women with hypertension	N = 10, 71.1 ± 5.5	Pre-exercise: 1.20 ± 0.30 60 min post-exercise: 1.30 ± 0.30	Pre-exercise: 15.00 ± 5.50 60 min post-exercise: 14.20 ± 4.60
Droguett, 2015 [51]	Cycle ergometer for 15 min at mild intensity	Healthy	N = 10, 66 ± 2	Pre-exercise: 1.20 ± 0.10 15 min post-exercise: 2.0 ± 0.1	
Droguett, 2015 [51]	Cycle ergometer for 15 min at moderate intensity	Healthy	N = 10, 66 ± 2	Pre-exercise: 1.20 ± 0.10 15 min post-exercise: 3.20 ± 0.10	
Machado-Vidotti, 2014 [52]	Upper limb resistance exercise	Healthy elderly men	N = 10, 65 ± 1.2	Pre-exercise: 3.0 ± 0.50 During exercise:16.0 ± 4.0	Pre-exercise: 20.00 ± 2.00 During exercise: 11.00 ± 3.00
Machado-Vidotti, 2014 [52]	Lower limb resistance exercise	Healthy elderly men	N = 10, 65 ± 1.2	Pre-exercise: 3.0 ± 1.0 During exercise: 5.0 ± 1.0	Pre-exercise: 29.0 ± 5.0 During exercise: 12.0 ± 2.0
Parks, 2022 [53]	Cycle ergometer, Passive Recovery	Healthy, moderately active individuals	N = 16, 23 ± 3	Pre-exercise: 3.9 ± 0.7 15 min post-exercise: 4.7 ± 1.0 30 min post-exercise: 5.5 ± 1.1 45 min post-exercise: 5.1 ± 1.1 60 min post-exercise: 4.7 ± 1.1	Pre-exercise: 4.50 ± 0.60 15 min post-exercise: 2.3 ± 0.9 30 min post-exercise: 2.8 ± 0.8 45 min post-exercise: 3.3 ± 0.8 60 min post-exercise: 3.60 ± 0.80
Parks, 2022 [53]	Cycle ergometer, Cool-Down	Healthy, moderately active individuals	N = 16, 23 ± 3	Pre-exercise: 3.5 ± 0.6 15 min post-exercise: 5.1 ± 0.9 30 min post-exercise: 5.2 ± 1.2 45 min post-exercise: 4.9 ± 1.1 60 min post-exercise: 4.4 ± 1.0	Pre-exercise: 4.50 ± 0.50 15 min post-exercise: 2.3 ± 0.7 45 min post-exercise: 3.4 ± 0.8 30 min post-exercise: 2.9 ± 0.8 60 min post-exercise: 3.60 ± 0.70
Sun, 2016 [54]	Treadmill exercise	Healthy Caucasian	N = 30, 24 ± 4	Pre-exercise: 192 ± 115 30 min post-exercise: 292 ± 229 60 min post-exercise: 233 ± 168	Pre-exercise: 64.84 ± 47.34 30 min post-exercise: 43.78 ± 36.41 60 min post-exercise: 59.84 ± 49.78
Sun, 2016 [54]	Treadmill exercise	Healthy Chinese	N = 32, 28 ± 4	Pre-exercise: 192 ± 194 30 min post-exercise: 517 ± 528 60 min post-exercise:296 ± 257	Pre-exercise: 69.47 ± 90.90 30 min post-exercise: 17.87 ± 15.42 60 min post-exercise: 40.30 ± 38.50
Marôco, 2024 [56]	HIIE 10 min	T2DM	N = 12, 67 ± 8	Pre-exercise: 3.2 ± 0.4 Immediate post-exercise: 3.2 ± 0.5	Not available
Marôco, 2024 [56]	HIIE 60 min	T2DM	N = 12, 67 ± 8	Pre-exercise: 3.2 ± 0.4 Immediate post-exercise: 3.1 ± 0.4	Not available
Marôco, 2024 [56]	MICE 10 min	T2DM	N = 12, 67 ± 8	Pre-exercise: 3.3 ± 0.3 Immediate post-exercise: 3.7 ± 0.3	Not available
Marôco, 2024 [56]	MICE 60 min	T2DM	N = 12, 67 ± 8	Pre-exercise: 3.3 ± 0.3 Immediate post-exercise: 4.3 ± 0.5	Not available
Bhati, 2019 [57]	A maximal exercise test	T2DM with CAN	N = 27, 51.7 ± 6.1	Pre-exercise: 1.5 ± 0.7 Immediate post-exercise: 1.4 ± 0.6	Pre-exercise: 24.6 ± 11.2 Immediate post-exercise: 10.7 ± 4.2
Bhati, 2019 [57]	A maximal exercise test	T2DM without CAN	N = 15, 50.2 ± 6.4	Pre-exercise: 1.7 ± 1.3 Immediate post-exercise: 1.2 ± 1.8	Pre-exercise: 30.7 ± 15.6 Immediate post-exercise: 19.6 ± 11.3
Colberg, 2014 [58]	Walk on a treadmill	Uncomplicated T2DM	N = 12, 58.7 ± 2.4	Pre-exercise: 2.9 ± 3.3 Immediate post-exercise: 2.6 ± 3.2	Pre-exercise: 1.4 ± 0.1 Immediate post-exercise: 1.3 ± 1.0
Colberg, 2014 [58]	Table tennis	Uncomplicated T2DM	N = 12, 58.7 ± 2.4	Pre-exercise: 2.4 ± 3.5 Immediate post-exercise: 2.8 ± 4.7	Pre-exercise: 1.4 ± 0.1 Immediate post-exercise: 1.3 ± 0.1
Grieco, 2014 [59]	Yoga	T2DM	N = 12, 54.9 ± 7.4	Not available	Pre-exercise: 1.5 ± 0.4 Immediate post-exercise: 1.5 ± 0.1

Abbreviations: CAN, cardiac autonomic neuropathy; LF/HF, the ratio between the absolute power of the high-frequency (HF) band (0.015–0.400 Hz) and the absolute power of the low-frequency (LF) band (0.040–0.150 Hz); HIIE, high-intensity interval exercise (consisting of 1 min exercise bouts at 90% of VO_2 Reserve_, alternated with 1 min active recovery bouts at 60% of VO_2 Reserve_ (1:1 ratio); MICE, moderate continuous exercise (protocol intensity was set at 60% of VO_2 Reserve_, and the duration was adjusted for each participant to achieve the targeted energy expenditure) [56]; PP, pulse pressure; RE, resistance exercise; RMSSD, the root mean square of successive heart beat interval differences.

**Table 2 medicina-60-01223-t002:** The estimate for RMSSD and LF/HF ratio by meta-regression.

Outcome	Moderator	*β*	95% CI	S.E.	Z Value	*p*-Value
RMSSD						
	Mean age	−0.10	−0.18–−0.03	0.04	−2.77	0.006
	Male, percent	−10.64	−17.19–−4.08	3.35	−3.18	0.002
	Post-exercise, mins	0.10	0.02–0.18	0.04	2.36	0.019
	Intercept	4.03	−4.20–12.26	4.20	0.96	0.337
LF/HF ratio						
	Health vs. chronic diseases	1.45	0.005–2.90	0.74	0.46	0.049
	Intercept	0.25	−0.82–1.32	0.55	1.97	0.64

Abbreviations: MD, mean difference; LF/HF, the ratio of the absolute power of the low-frequency band (0.04–0.15 Hz) to the absolute power of the high-frequency band; RMSSD, the root mean square of successive differences between normal heartbeats.

## Data Availability

No new data were created or analyzed in this study. Data sharing is not applicable to this article.

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
