# Peer review of "The Impact on Autonomic Nervous System Activity during and Following Exercise in Adults: A Meta-Regression Study and Trial Sequential Analysis"

_medicina, 2024, doi:10.3390/medicina60081223_

Round 1

Reviewer 1 Report

Comments and Suggestions for Authors

·       In the abstract, last line, do the authors mean health conditions? It is stated as healthy.

·       Since this systematic review concerns the autonomic nervous system activity concerning exercise, it is important to briefly mention the autonomous nervous system and its role and also to define various terminologies used in this manuscript. The authors have used exercise, physical activity, training, etc. It seems a bit difficult to understand what is what and if it is important to know pr definition of what these are so that one can link those to the autonomic nervous system activity. Please elaborate.

·       In the search word, this issue is also seen. The authors have used terminologies related to all of these. If this has been the am, please add and elaborate further on whether these separate words and terminologies are different and can influence the results or not.

·       What has been the aim of this meta-analysis? The authors have compared several places of their study with already available systematic reviews or previously reported results. The authors must add the novelty of this review and analysis to justify why we needed this in addition to the already existing reviews in the field. In other words, please add clearly what this review adds.

·       Another important point that the authors are encouraged to add is the clinical implications of these findings. How can we use these in the clinic or research? Please add and give a translational perspective to the findings of this study.

·       Please also add why among more than 3000 screened articles only 10 were selected, what were the exclusion reasons for so many articles, and if these were not relevant why they were captured by the search strategy in the search process?

·       The authors have searched Mdline and PubMed. What is the difference between these two and is that necessary to search both or one can be sufficient among these two particular databases

·       In the conclusion, please add that in light of the limitations of this study the findings have been concluded that way.

·       Please add the sources of potential bias in this review.

Author Response

Dear Reviewer, 
Thank you for your kind review of our manuscript.
The attached PDF file is our sincere reply to your kind suggestions.

Reviewer 2 Report

Comments and Suggestions for Authors

Here are my comments and suggestions:

  1. Relatively small number of included studies (10) and total participants (292), which limits generalizability.
  2. High heterogeneity among included studies in terms of exercise protocols, participant characteristics, and measurement timing.
  3. Potential publication bias for RMSSD results, as indicated by Egger's test.
  4. Reliance on LF/HF ratio as a measure of sympathetic activity, which is controversial and may not accurately reflect sympathetic tone.
  5. Limited analysis of exercise intensity and duration effects due to inconsistent reporting across studies.
  6. Lack of long-term follow-up data to assess chronic adaptations to exercise.
  7. No assessment of exercise adherence or participant characteristics like fitness level that could influence results.
  8. Limited exploration of potential mechanisms underlying observed autonomic changes.

Suggestions for improvement:

  1. Conduct subgroup analyses based on exercise modality, intensity, and duration if data permits.
  2. Include a more detailed assessment of study quality and risk of bias.
  3. Discuss limitations of using LF/HF ratio as a sympathetic measure and consider alternative indices if available.
  4. Explore the clinical significance of observed autonomic changes, particularly for participants with chronic conditions.
  5. Provide more specific recommendations for future research to address gaps identified in the review.
  6. Consider including a brief overview of autonomic nervous system physiology for readers less familiar with the topic.
  7. Expand on potential mechanisms linking exercise to autonomic function changes.

Overall, this manuscript provides a valuable contribution to understanding the acute effects of exercise on autonomic function, but there are opportunities to strengthen the analysis and enhance its clinical relevance

Author Response

(The authors gave the same response as above.)

Round 2

Reviewer 2 Report

Comments and Suggestions for Authors

The authors responded to my comments well. Thank you.